# Heterogeneous Architecture Search Approach within Adversarial Dynamic Defense Framework

**Written by Qi Peng,**[1] **Ruoxi Qin,**[1] **Wenlin Liu,**[2] **Libin Hou,** [1] **Bin Yan,** [1] **Linyuan Wang**[1] [*]

[1]Henan Key Laboratory of Imaging and Intelligent Processing, PLA Strategy Support Force Information Engineering University
[2]Department of Electronic Engineering and Information Science, University of Science and Technology of China (USTC)
pqmailwiki@163.com, wanglinyuanwly@163.com

## Abstract

Recent advances in adversarial attacks uncover the intrinsic vulnerability of modern deep neural networks (DNNs). To address this issue, various methods have been proposed to design network architectures that are robust to one particular type of adversarial attack. Recent research leverages the concept of dynamic defense framework (DDF) based on stochastic ensemble model for boosting the robustness of a DNN ensemble against such adversarial attacks. There is a need to enhance the diversity and gradient variations of the ensemble but stuck with the lack of efficient networks. In this paper, we propose a heterogeneous architecture searching method based on NAS. Our method encourages heterogeneous networks, such that networks further improve diversity for ensemble, and thus, boost the adversarial robustness of DDF. Experimental results suggest that the diversity existing among the family of heterogeneous networks does restrain the transferability of the adversarial sample, and achieve superior performance when evaluating the robustness on the ASR-vs-distortion benchmark in different attack environments.

## Introduction

Deep neural network shows excellent performance in the application on standard classification tasks. Nevertheless, (Szegedy et al. 2013) proposed that even minor disturbances can mistakenly lead to the wrong prediction of state-of-the-art (SOTA) classifiers. With the sustainable development of adversarial attack, the defenders have proposed many strategies to enhance effect against the treat, most notably adversarial training (Goodfellow, Shlens, and Szegedy 2014; Madry et al. 2017). Adversarial training minimizes the loss of a DNN model on online-generated adversarial examples against the sample itself at each training step. However, adversarial training leads to the degenerative generalization ability of DNN (Bai et al. 2021). Thus, adversarially trained models can be conquered easily by newer or different types of attacks (Kang et al. 2019). Such terrible generalization to other attacks prominently degrades the reliability of adversarial training.

Adversarial training, which are robust to a specific attack, are particularly passive in the arm race. On the contrary, model ensemble is an effective defense strategy in practice to construct an ensemble of individual models (Kurakin et al. 2018). Researchers build upon the dynamic defense framework (DDF) based on the stochastic model ensemble method (Qin et al. 2021). The DDF method employs network architecture and smoothing parameters as ensemble attributes, and dynamically change attribute-based ensemble model before every inference prediction request. To ensure an extensive range of gradient directions and meet the diverse needs of the stochastic ensemble, each model in the ensemble needs to have widely varied gradient information. Other studies also pointed out the crucial role of this diversity in ensemble defense methods (Pang et al. 2019) and proposed methods (D'Angelo and Fortuin 2021; Wang et al. 2019) to maintain diversity among the members and transform the maximum a posteriori inference into proper Bayesian inference.

Neural architecture search (NAS) (Baker et al. 2016; Zoph and Le 2016) affords the opportunity to discover diversified networks to produce an improved ensemble result. As the goal of NAS is to find various network architectures, the searched networks can replace the design implemented by human experts. In this paper, we use NAS method to discover a family of heterogeneous networks, which improve the diversity needed for dynamic defense framework to achieve better performance on ensemble robustness. Through this method, we discover heterogeneous networks that introduce model size and quantity as variable attributes for diversity, which achieves unpredictable gradient of ensemble. The unpredictability of gradient makes it impossible for attackers to directly implement white-box attacks, and the inefficient universality of the adversarial samples make the attacker helpless.

## Method

### Dynamic defense framework

To begin with, we need to understand the concept of dynamic defense framework to capture the following idea of problem-solving. The dynamic defense framework combines heterogeneous redundant model collection and stochastic ensemble strategy.

**Heterogeneous redundant model collection.** According to the moving target defense (MTD) method (Jajodia et al. 2011), the basic requirements of the robust-control model

---

[*]Corresponding author.

are randomness, diversity, and dynamics. As we can see, the heterogeneous redundant model collection is a good demonstration of dynamic defense framework. The heterogeneity of the model structure leads to a large range of gradient directions for the ensemble. Furthermore, the redundancy of the model library not only performs well in classification but also makes the model structure diverse.

**Stochastic ensemble method.** The stochastic ensemble method(SEM) randomness is reflected in the randomization of the model attributes $\theta$. The ensemble model $f_{ens}$ containing K numbers of models is defined as follows:

$$f_{ens}(x,\theta) = \sum_{k=1}^{K} f(x,\theta_k). \tag{1}$$

Specifically, when the SEM is queried multiple times for gradient or output information, the ensemble is achieved through an utterly stochastic choice of the ensemble attributes $\theta$ in each query, which can effectively shield model gradient. Before each rational prediction request, the adversarial samples generated by the white-box attack will be confused by the ensemble model based on the dynamic defense of variable attributes—the attack results in different update directions of gradients.

In short, dynamic defense framework is instrumental in confusing the attacker's strategy. It is achieved by randomly selecting ensemble attributes and reflecting the unconfirmed probabilistic gradient information during each iteration of the ensemble.

### Searching heterogeneous model based on NAS

The research of DDF generates diversity by the designed networks, which is still preliminary work. Broader diversity is needed, we have been explored a wider range of model diversity to make it more relevant.

**Define proper search space**. For our complex task, we employ the cell-based search spaces to remains a solid searching capability while reducing the burden of computation. Cell-based NAS aims to search for cells as the fundamental part of the whole architecture(Bender et al. 2018). The architecture of each cell is an association of operations chosen from a predefined operation space. We have reduced the number of candidate operations to 3*3 separable convolution, identity, and zero, which can still maintain the diversity and possibility of solid search space (Xie et al. 2019). It still maintains $4^{14} - 1 \approx 10^8$ viable cells. Moreover, the diverse hyperparameters of the network architecture plays crucial role in the searching progress. Hyperparameters, like connection method, the number of convolutional layer and channels used for feature extraction, define which network architectures can be searched by the NAS. Such a scale of search space can improve the dynamic property of the ensemble. It effectively expands the diversity of the model collection.

**Set diversity metrics of ensemble**. We choose diverse hyperparameters of the network architecture and quantity of ensemble networks as variable attributes of DDF. These hyperparameters plays crucial role in the searching progress.

For example, architectural template, the number of convolutional layer and channels used for feature extraction, separate searched networks by NAS to disparate architectures. Specially, we set connection mode as a variate of ensemble. For cell-based settings, the possibilities we can choose from the convolution operations is 14. As the dense connection mode benefits the robustness of the network(Guo et al. 2020), we choose two types of connections. On one hand, we limit the number of connections greater than 11 to maintain the robustness of network, part of the free-connection models show in Table1. On the other hand, we release the cell-based constraint to further improve the diversity of the model, we put some of the dense-connection models in Table2.

Table 1: Portion of free-connection models from heterogeneous model library

| Channels | Layers | Connect | Clean | PGD-20 |
|---|---|---|---|---|
| 36 | 20 | free | 82.7 | 52.6 |
| 36 | 21 | free | 79.5 | 48 |
| 36 | 22 | free | 79 | 48.2 |
| 64 | 10 | free | 81.2 | 49.1 |
| 64 | 14 | free | 82.9 | 50.3 |
| 64 | 15 | free | 76.7 | 48.9 |
| 64 | 18 | free | 79 | 48.5 |

Table 2: Portion of dense-connection models from heterogeneous model library

| Channels | Layers | Connect | Clean | PGD-20 |
|---|---|---|---|---|
| 36 | 10 | 11-14 | 78.3 | 48.2 |
| 36 | 14 | 11-14 | 78.9 | 47.7 |
| 36 | 18 | 11-14 | 77.5 | 49.5 |
| 36 | 20 | 11-14 | 78.5 | 49.1 |
| 36 | 20 | 11-14 | 83.5 | 52.2 |
| 36 | 28 | 11-14 | 76.8 | 46.7 |
| 36 | 36 | 11-14 | 83.9 | 53.5 |
| 64 | 18 | 11-14 | 78 | 50.2 |
| 64 | 20 | 11-14 | 80 | 50.7 |
| 64 | 22 | 11-14 | 80.2 | 50.6 |
| 64 | 26 | 11-14 | 80 | 47.2 |
| 64 | 33 | 11-14 | 85.7 | 49.6 |
| 64 | 36 | 11-14 | 80.7 | 49.1 |

**Combine heterogeneity and robustness.** We implement the idea about searching for heterogeneous and robust architectures simultaneously in NAS search process without changing the weights of networks. According to (Guo et al. 2020; Tang et al. 2021), network architecture, especially model size and families, has a significant impact on model robustness. Our proposed method applies generated adversarial sample of each sub-network by PGD attack (Madry et al. 2017) to evaluate the sub-network itself during NAS searching progress. The problem of defending against bounded adversarial perturbations can be formulated as fol-

lows:

$$\min_{\theta} E_{(x,y)\sim\mathcal{D}} \left[ \max_{x'\in S} \mathcal{L}\left(y, M\left(x';\theta\right)\right) \right] \quad (2)$$

Where $S = \|x - x'\|_p < \varepsilon$, which defines the set of permissible disturbance inputs within the $L_p$ distance, where M denotes the candidate network and D represents the data. In our study, we mainly use PGD adversarial samples within the $L_2$ norm to obtain robust networks with different architectures.

## Selecting variable attributes Dynamically for robust ensemble

For model ensemble, the gradient change is determined by the variation of dynamic attributes and gradient of each model. We ultimately select four properties of the model as variable attributes, which are the number of models, the depth of models, the width of models, and the density of connections. In each prediction, we randomly choose variable attributes when predicts output information multiple times.

As we contain the model diversity into the ensemble attributes, each iteration of the ensemble causes transformation for the direction of gradient. Our experiments show that the randomness has a specific effect to confuse the attackers. And the heterogeneous models further improves the robustness performance and solves the vulnerability of the ensemble under attack. We provide the pseudo code of our heterogeneous architecture search algorithm for DDF as Algorithm 1.

---

Algorithm 1: Heterogeneous architecture search algorithm for DDF

---

**Input**: Model library capacity *M*, model variable attributes $\theta(l, c, d)$, network parameter $\alpha$, NAS total iterations *I*, ensemble total iterations *J*.

1: **for** m=0...*M* **do**
2:    Set $\theta(l, c, d)$).
3:    Search Super-net *G* in accordance with $\theta(l, c, d)$.
4:    **for** i=0...*I* **do**
5:       Train the randomly sampled sub-model *S* with PGD adversarial sample.
6:       Update $\alpha_i$ by SGD.
7:    **end for**
8: **end for**
9: **for** j=0...*J* **do**
10:   Randomly select the number of ensemble models *N*.
11:   Perform ensemble and inference prediction.
12: **end for**

---

# Experiment and Results

In this section, we evaluate our proposed method for its adversarial robustness by ASR-vs-distortion curve (Dong et al. 2020). Unlike the traditional point estimation method, ASR-vs-distortion curve is different from the validation of the robust radius, but the experiment of evaluating the robustness of the actual attack environment.

## Experimental setup

In this subsection, we would demonstrate implementation details of our experiments. As described in Sec. 2, we choose the number of models, the depth of models, the width of models, and the density of connections as the variable attributes of ensemble. The evaluation is only on CIFAR-10.

We can find the minimum perturbations of attacks by counting the number of the adversarial examples. Not just single perturbation, but we calculate the accuracy or attack success rate for all possible values of $\varepsilon$, which represent the perturbation budget. Then we perform a binary search on $\varepsilon$ to find its accurate minimum value that enables the generated adversarial example to fulfill the adversary's goal. We'll show the accuracy (attack success rate) vs. perturbation budget curve, which can reveal a global understanding of the robustness of a classifier. To assess full-scale robustness, we set the search step to 10, and binary search step to 20 with $L_2$ norm adversarial attacks.

In this paper, the DNN model under attack is used as the baseline model A for contrast. Model B presents DNN models within DDF, model C presents SOTA architecture method Robnet(Guo et al. 2020) concerning the impact of architecture to robustness, model D presents single NAS model, E presents NAS models within DDF that ensemble the same quantity of sub-models as B. Specially, F presents the same model with E, but we sample the results of 10 disparate models on all distortion value under evaluation. These sampled models cope with adversarial samples generated by diverse ensemble model. Through observation for the attack success rate, we can have a explicit judgement whether the ensemble model could cutoff transferability of adversarial samples.In order to control the potential empirical model risk, the abstention threshold $\alpha$ is set to 0.3 for bounding the acceptable probability of returning an incorrect prediction.

## Robustness analysis

The result on ensemble models can be observed with PGD attack, which is commonly employed for evaluation because of its attack effectiveness in white-box settings. Adversarial examples are crafted by untargeted method.

As shown in Figure 1, the ensemble model of the DNN is extremely vulnerable under white-box attacks, against which the attack success rate is higher when the perturbation stays low. For the DNN ensemble method B and NAS ensemble method E, F, we can see NAS ensemble model,with the same quantity of sub-models, outperforms each case of the other methods with higher demand of perturbation budget. On the observation of C and F, attacker overcome Robnet under lower perturbation values. Heterogeneous architecture search method improves the robust performance of ensemble model by enhancing the diversity of sub-models, and solves the vulnerability of ensemble under white-box attacks. Compare the performance of D with F, the DDF does improve the randomness of the gradient, which has a certain effect on the confusion of the gradient direction.

## Transferability analysis

In this sub-section, we show that our method can ease the transferability of adversarial examples for the ensemble. As

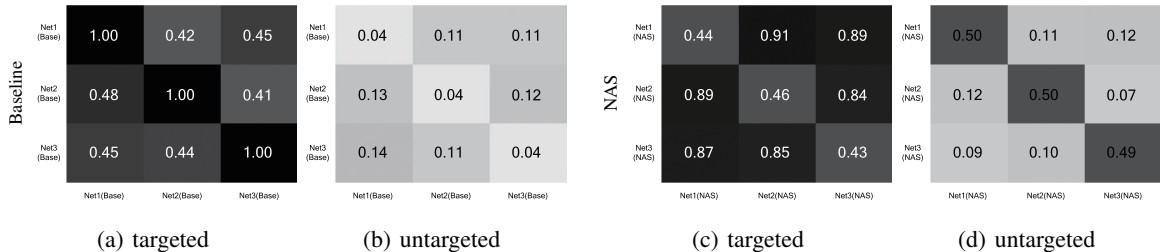

|  | (a) targeted | | (b) untargeted | | (c) targeted | | (d) untargeted |
|---|---|---|---|---|---|---|---|

Figure 1: Adversarial transferability among individual models on CIFAR-10. We take the classification accuracy of the models on adversarial examples for untargeted mode, and success rate of mislead the models to specific target label for targeted mode.

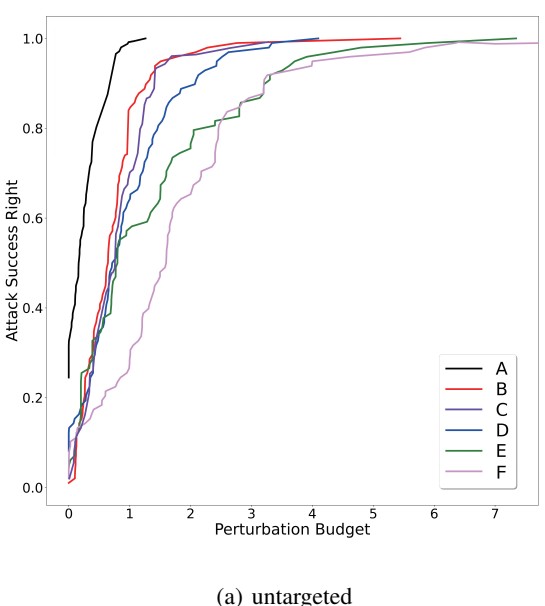

(a) untargeted

Figure 2: The ASR-vs-distortion curves for PGD transfer-based untargeted white-box attacks. A: single DNN model under attack. B: the ensemble DNN model under attack. C: Robnet under attack. D: single NAS model under attack. E: the NAS ensemble model without sampling. F: the NAS ensemble model with sampling.

shown in Figure 1, F has a better robust performances than E, this reveals that adversarial sample crafted by one of our heterogeneous models have poor performance on the rest models.

In addition, for another transferability evaluation, we use random network as the substitute model to craft adversarial examples and feed to another two networks as the original model. We apply PGD as the attack method, which are the most commonly used attacks in the black-box setting (Kurakin et al. 2018). The perturbation parameters for both attacks are set to be 20/255. We test untargeted mode and targeted mode. The evaluation standard is different for two modes. We take the classification accuracy of the models on adversarial examples for untargeted mode, and success rate of fooling the models to predict specific target label for targeted mode. The Figure 2 presents the pair-wise transferability with three models of NAS method and DNNs. the adversarial samples effect on the model crafting them, but the disparate model can greatly hinder the transferability among heterogeneous models in both modes. This phenomenon demonstrates the heterogeneity of our model library, which can further enhance ensemble robustness.

## Conclusion

This study set out to improve the heterogeneity and diversity of dynamic defense framework generalized by DNNs. In this paper, we take advantage of the diversity of network structures discovered by NAS, due to the complexity of ample search space and the maneuverability, which drives a heterogeneous model library. Experimental results show that the diversity of searched models does enhance the performance of ensemble model and prevent the transferability of adversarial samples under different attack environments. To further improve the robustness of dynamic defense framework and effectively prevent the transfer of adversarial sample on different models, the evaluation process of NAS needs a better expression of model diversity.

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
