# OpenReview forum: "Heterogeneous Architecture Search Approach within Adversarial Dynamic Defense Framework"
_AAAI.org/2022/Workshop/AdvML — AAAI-22 AdvML Workshop ShortPaper_

### Official Review · Reviewer_bypE · 2021-11-29

**Rating:** 6
**Confidence:** 4

**Review:**

This paper proposes a heterogeneous architecture searching method based on NAS. The proposed method encourages heterogeneous networks, such that networks further improve diversity for ensemble, and thus, boost the adversarial robustness of DDF. The empirical studies are insightful.

Main concerns:

1. There are works in the Bayesian deep learning community that try to diversify the deep ensemble in principle, see [1, 2]. Can the proposed approach be closely connected with them to improve the novelty?

2. The current empirical studies are not so thorough. It is better to show this approach can achieve a SOTA performance.

[1] Function Space Particle Optimization for Bayesian Neural Networks. Wang et al., ICLR 2019.
[2] D'Angelo, Francesco, and Vincent Fortuin. "Repulsive Deep Ensembles are Bayesian." NeurIPS 2021

---

### Decision · Program_Chairs · 2021-12-01

**Decision:**

Accept (Short Paper)

**Comment:**

The reviewer agrees to accept this paper. Please address the reviewer's comment in the camera-ready version.